# Assessment of Intestinal Ischemia–Reperfusion Injury Using Diffuse Reflectance VIS-NIR Spectroscopy and Histology

**DOI:** 10.3390/s22239111

**Published:** 2022-11-24

**Authors:** Jie Hou, Siri Schøne Ness, Jon Tschudi, Marion O’Farrell, Rune Veddegjerde, Ørjan Grøttem Martinsen, Tor Inge Tønnessen, Runar Strand-Amundsen

**Affiliations:** 1Department of Physics, University of Oslo, Sem Sælands vei 24, 0371 Oslo, Norway; 2Department of Clinical and Biomedical Engineering, Oslo University Hospital, 0424 Oslo, Norway; 3Department of Pathology, The Norwegian Radium Hospital, Oslo University Hospital, Ullernchausseen 70, 0379 Oslo, Norway; 4SINTEF AS, Smart Sensors and Microsystems, Forskningsveien 1, 0373 Oslo, Norway; 5Sensocure AS, Langmyra 11, 3185 Skoppum, Norway; 6Department of Emergencies and Critical Care, Oslo University Hospital, 0424 Oslo, Norway; 7Institute of Clinical Medicine, University of Oslo, 0318 Oslo, Norway

**Keywords:** small intestine, VIS-NIR spectroscopy, ischemia–reperfusion injury, histopathology

## Abstract

A porcine model was used to investigate the feasibility of using VIS-NIR spectroscopy to differentiate between degrees of ischemia–reperfusion injury in the small intestine. Ten pigs were used in this study and four segments were created in the small intestine of each pig: (1) control, (2) full arterial and venous mesenteric occlusion for 8 h, (3) arterial and venous mesenteric occlusion for 2 h followed by reperfusion for 6 h, and (4) arterial and venous mesenteric occlusion for 4 h followed by reperfusion for 4 h. Two models were built using partial least square discriminant analysis. The first model was able to differentiate between the control, ischemic, and reperfused intestinal segments with an average accuracy of 99.2% with 10-fold cross-validation, and the second model was able to discriminate between the viable versus non-viable intestinal segments with an average accuracy of 96.0% using 10-fold cross-validation. Moreover, histopathology was used to investigate the borderline between viable and non-viable intestinal segments. The VIS-NIR spectroscopy method together with a PLS-DA model showed promising results and appears to be well-suited as a potentially real-time intraoperative method for assessing intestinal ischemia–reperfusion injury, due to its easy-to-use and non-invasive nature.

## 1. Introduction

Insufficient blood supply (ischemia) to the small intestine often requires invasive surgery to restore the supply. Early detection and correction of the ischemic intestinal segment is the most important factor for patient survival [1]. The standard clinical method for assessing small intestinal ischemia is based on a subjective visual and tactile evaluation of the tissue, such as observing changes in color, the occurrence of peristalsis or a change in the thickness of the intestinal wall [1,2,3]. This method has low initial accuracy, which can often lead to a second-look surgery, where the questionable segments are re-examined within 24–72 h after the original surgery. It would be a significant improvement to patient treatment if a more reliable method to detect irreversible ischemia could be established. This study is part of the work that explores more reliable, objective methods for gastrointestinal surgeons.

Over the years, various experimental approaches have been investigated to help increase the accuracy when assessing intestinal viability, including fluorescein and laser Doppler flowmetry [4,5,6]. The fluorescein technique has a low detection threshold and is affected by tissue autofluorescence, which will generate background noise and false signal [7]. The laser Doppler flowmetry is sensitive to motion artifacts and blood pressure changes [8], and large standard deviations in the flow parameter make continuous measurement necessary [8,9,10]. Moreover, intraoperative Doppler ultrasound has been applied to determine the adequacy of blood supply in the small intestine [11]. High false-negative and false-positive rates were observed using Doppler ultrasonography [12]. Microdialysis as a minimally invasive approach has been investigated by Deeba et al. [13], where they monitored the glucose and lactate concentrations. Implanting the microdialysis catheter near the bowel segment for postoperative monitoring was suggested by the authors. More recently, Strand-Amundsen et al. [14] further investigated the feasibility of microdialysis by monitoring the concentration change in the intraluminal lactate and glycerol. They were unable to find a sufficient level of association between intestinal viability and the metabolic markers, to conclude the clinical relevance of microdialysis with respect to the assessment of tissue viability. Most of the above-mentioned experimental approaches focus on blood flow. There are only a few studies that focus on the viability outcome of the ischemia–reperfusion injured intestine. Recently, Strand-Amundsen et al. [15] used bioimpedance measurements together with machine learning algorithms (LSTM-RNN) to more accurately assess intestinal viability in a pig model. Some years later, Hou et al. [16] used bioimpedance measurements on human intestine ex vivo, reporting an association between the $P_{y}$ value from the bioimpedance data and the viability of the small intestine. Moreover, Hou et al. [3] investigated the use of dielectric relaxation spectroscopy together with machine learning methods (LSTM and CNN) to assess small intestinal viability.

Visible and Near-Infrared (VIS-NIR) spectroscopy can potentially be used to assess intestinal ischemia–reperfusion injury objectively, without the need for a contrast agent [5,17,18]. As the blood flow is blocked during ischemia, the oxygenated haemoglobin levels will decrease, whereas deoxygenated haemoglobin levels increase, and this will be reversed during the reperfusion phase. Oxygenated haemoglobin and deoxygenated haemoglobin have spectral features in the wavelength range from 500–1000 nm. Therefore, information related to ischemia–reperfusion injury should be present in the VIS-NIR reflectance spectrum of the intestinal tissue. Pulse oximetry is a non-invasive method that compares the signals at 660 nm and 940 nm to monitor blood oxygenation (SpO_2_). Traditional pulse oximetry is dependent on the inherent alternating component of the signal, which is present when there is an active and strong arterial pulse in the tissue, to correctly calculate SpO_2_. VIS-NIR diffuse reflectance spectroscopy, on the other hand, is not dependent on the pulsative signal and can also work in conditions with little or no pulse, such as ischemia or reperfusion.

VIS-NIR spectroscopy has been investigated in several earlier studies. Sowa et al. [1] used near-infrared spectroscopy to detect intestinal ischemia in a porcine model in the wavelength range 500–1000 nm. They compared the reflectance spectra from control (no vascular manipulation), ischemic (arterial/venous occlusion up to 90 min) and reperfused (up to 60 min) intestinal segments. Furthermore, a classification model built using PLS-DA was used to differentiate between the control and ischemic segments, where the overall classification accuracy was 89%. Karakas et al. [5] investigated the viability of the intestine in a rat model using diffuse reflectance spectroscopy in the wavelength range 450–750 nm, where they focused on the ratio of the absorption between wavelengths at 560 nm and at 577 nm. Furthermore, they correlated this ratio to the histopathological grading with an 8-point Park/Chiu score.

This study investigates the feasibility of VIS-NIR diffuse reflectance spectroscopy in assessing the small intestinal ischemia–reperfusion injury. Compared to the above-mentioned studies, the contribution of this study includes: (1) The method is non-invasive and does not require applying any active signal or agent to the patient (can simplify the approval process of a medical device). (2) With the DR-probe, the measurement field of view is large and the measurement site is isolated from ambient room light, which gives high reproducibility of the experiment/data compared to “single point” measurements. It is much simpler and faster to scan/measure a large part of the small intestine with this probe, compared to the previously reported “point” measurements. (3) Reflectance data over a wide wavelength range together with the PLS-DA model provide high prediction accuracy in viability assessment, which is highly clinically relevant. Different models were built for different datasets. By comparing them, a better understanding of the spectral wavelength response during ischemia–reperfusion was gained. An accurate and objective method of detecting viable versus non-viable intestinal tissue would provide surgeons with a decision support tool for resection margins. This could potentially reduce the need for a second-look surgery, contributing to a better prognosis for the patient.

## 2. Methods

### 2.1. Animals and Experimental Design

Ten Norwegian Landrace pigs were included in this study, with a weight range of 48–54 kg, and three were female. The pigs were normal, healthy pigs and were screened by a professional pig breeder, to avoid disease. Food was withheld 12 h prior to surgery. The same experimental protocol was used as described by Strand-Amundsen et al. and Hou et al. [2,3,19]. For each pig, four random segments of jejunum were selected for the four cases of the protocol:**Case 1**: Control (Reference)—8 h of normal perfusion;**Case 2**: Ischemia—8 h of warm full ischemia;**Case 3**: 2 h of ischemia followed by 6 h of reperfusion;**Case 4**: 4 h of ischemia followed by 4 h of reperfusion.

Ischemia was induced by clamping the arteries and veins of the jejunal mesentery of the selected segments, resulting in a 30 cm central zone of warm ischemia and two surrounding edge zones of marginal tissue hypoxia [2,19]. The time durations for segment Case 3 and segment Case 4 were chosen based on histological analysis, which predicts that irreversible injury is estimated to occur after around 2 h of ischemia. In-vivo spectroscopic measurements were conducted hourly over an 8-hour period. All samples received the same treatment, while ischemic exposure varied dependent on the protocol. Prior to implementing the protocol, a couple of pigs were selected for histological analysis, where biopsy samples were taken from all the aforementioned cases, in addition to a case with 3 h of ischemia and 5 h of reperfusion. This was accomplished in order to follow the time sequence of the ischemia changes more closely. After the experiment, the animals were euthanized by a lethal dose of potassium chloride (100 mmol). The experiment was approved by the Norwegian Food Safety Authority (NFSA) and conducted in accordance with national animal welfare guidelines.

### 2.2. Visible and Near-Infrared Diffuse Reflectance Spectroscopy Setup

Visible and near-infrared diffuse reflectance spectroscopy uses light and matter interaction to detect and determine the composition of organic substances. Light interacts with various molecules (e.g., water, fat, hemoglobin, lactate, glucose) in the tissue and the molecules behave and absorb visible and near-infrared light differently. The amount of light absorbed is linearly proportional to the concentration of that particular molecule, revealing both qualitative and quantitative information about the pathological process under investigation. Histopathological correlation with diffuse reflectance spectroscopy of tissue can be used to build a prediction model to detect and distinguish different tissue conditions [20].

The VIS-NIR measurements were performed using a handheld USB 2000+VIS-NIR (USB2+H00178) spectrometer (Ocean Insight, Orlando, FL, USA). It was operated in reflectance mode in the wavelength range 475–1100 nm with a 25 μ m slit for optical resolution of 1.5 nm (full width at half maximum). The spectrometer was connected to the QP600-2-VIS-NIR fiber (Ocean Insight, Orlando, FL, USA) with wavelength range 400–2100 nm and the TC-DR probe (Ocean Insight, Orlando, FL, USA) which is a 45° diffuse reflectance probe with integrated tungsten halogen light source (6 watts). The TC-DR probe has a 40 mm stand-off block ensuring consistent focal length and the field of view is 24 mm × 47 mm. Before acquiring the time series of in vivo diffuse reflectance data, two spectra were collected for the system calibration. The first one was the reference spectrum (R(λ)) measured using a spectralon reflectance standard with high reflectivity while the light source was turned on. The second spectrum acquired was the background spectrum (D(λ)) taken while the light source was turned off. The percentage reflectance as a function of wavelength (T(λ)) was calculated as follows:(1)$$\%T\left(\lambda \right)=\frac{S(\lambda )-D(\lambda )}{R(\lambda )-D(\lambda )}\times 100\%$$
where S(λ) is the sample intensity as a function of wavelength λ. All of the spectra were acquired with the OceanView software. Boxcar smoothing was set to 5, averaging was set to 50 scans, and each scan was taken with 3740 μ s integration time. The system was re-calibrated prior to each hourly measurement to maintain the accuracy of the system and minimize the drift error. Prior to each measurement, the small intestinal segment was gently wiped with a sterile cloth to avoid body fluid oozing under the area covered by the TC-DR probe and any possible contamination. After each hourly measurement, the abdomen was closed using several Backhaus towel clamps to minimize fluid and heat losses. Every hour, each intestinal segment was measured three times, at three different locations. Figure 1 shows the experimental setup for in vivo measurements.

### 2.3. Data Analysis and Classification

To minimize variations in the diffuse reflectance spectra induced by light scattering (due to, for instance, textural differences) and the different positions of each measurement, the spectra were normalized by standard normal variate (SNV). For each reflectance spectrum, the mean value was subtracted and then divided by the standard deviation of the spectrum, to transform the data to have mean μ = 0 and standard deviation σ = 1 using Equation (Equation 2):(2)$$Z=\frac{X-\mu }{\sigma }$$

A first-order derivative was applied to the standardized data to remove the baselines. Savitzky–Golay (SG) polynomial derivative filters were applied during derivation (the length of the filter window was set to 25). It applies a smoothing function to the spectra before calculating the derivative, to decrease the detrimental effect on the signal-to-noise ratio that conventional finite-difference derivatives would have [21].

Partial least squares-discriminant analysis (PLS-DA) was used to classify the measured diffuse reflectance spectra in the wavelength range 500–1000 nm. Data from different tissue conditions were labelled and two models were built; one performed classification of three classes (control, ischemic, and reperfused intestinal tissue); the other performed classification of two classes (viable and non-viable intestinal tissue). The viable and non-viable classes were determined based on the histopathological analysis. The viable class included data from the control case, ischemia ≤ 2 h, and reperfusion data from those which had previously been ischemic for ≤2 h. The rest of the data (ischemia > 2 h and reperfusion data from those which had previously been ischemic for >2 h) were classified as non-viable as Table 1 shows.

Furthermore, two datasets were tested for each of the models. One dataset was the SNV transformed and the other dataset was first SNV transformed and then first order differentiated. In addition, 10-fold cross-validation was used to validate the models, and the average scores (accuracy, sensitivity and specificity) with standard deviation are reported. The accuracy, sensitivity and specificity scores are defined as:$$\mathrm{Accuracy}=\frac{\mathrm{TP}+\mathrm{TN}}{\mathrm{TP}+\mathrm{FP}+\mathrm{FN}+\mathrm{TN}}$$
$$\mathrm{Sensitivity}=\frac{\mathrm{TP}}{\mathrm{TP}+\mathrm{FN}}\;\mathrm{Specificity}=\frac{\mathrm{TN}}{\mathrm{TN}+\mathrm{FP}}$$
where the model performance depends on True Positives (TP), True Negatives (TN), False Positives (FP) and False Negatives (FN).

During 10-fold cross-validation, for each iteration, data from 9 pigs (9 separate experiments) were used as training data, and data from one pig were used as test data. In this way, we obtained well-validated models where the training data and the test data were fully independent of each other.

PLS-DA is generally well suited for spectroscopic data, where the number of wavelengths is higher than the number of samples and collinearity exists among the variables. During modelling, PLS attempts to find orthogonal linear combinations of latent variables that explain the variability in the spectroscopic data, while being correlated to the response variable. By analyzing the coefficients, we can study the correlations between wavelength and the outcome variable. This allows us to investigate the most important wavelengths to distinguish intestinal tissue with respect to viability. Each input spectrum is given a class number. For the classification of two classes, spectra from viable intestinal tissue were assigned to class “0” and non-viable to class “1”. For the classification of three classes, spectra from the control intestinal segment were assigned to class “0”, ischemic intestinal segments to class “1” and reperfused segments to class “2”. The PLS-DA models were built in Python using the Scikit-learn library.

To find the optimal number of latent variables in the PLS-DA model, a grid search was performed for a range of 2–12 latent variables. For each loop, PLS-DA was run for a certain number of latent variables and obtained an average accuracy score. The optimal number of variables in the model was determined after cross-validation, by evaluating the accuracy score between the true class and the predicted class using the accuracy score from the 10-fold cross-validation. The optimal number of variables was found to be 9, the overall accuracy score started to decrease when a higher number of variables was used.

### 2.4. Histopathological Examination

To assess the histopathological changes in the tissue, full-thickness biopsies (20 mm × 5 mm) were taken from control, ischemic and reperfused intestinal segments. Specimens from the intestines were fixed overnight in buffered 10% formalin, dehydrated in a graded series of ethanol, and embedded in paraffin. 4-μ m-thick sections were cut and then stained with haematoxylin and eosin (H&E). Stained sections were examined by two pathologists simultaneously, and by consensus, each section was classified. The criteria of Swerdlow [22] were used since they are generally accepted and have also been used in previous work by our group [14].


**Modified Swerdlow classification of ischemic changes in the small intestinal wall:**
**Grade 0**: No pathological changes;**Grade 1**: Focal loss of surface epithelium in the mucosa;**Grade 2**: Mucosal infarction, with extensive loss of surface epithelium and areas with substance loss in the mucosal lamina propria. Sparing of basal parts of glands, and of the lamina muscularis mucoae;**Grade 3**: Complete mucosal necrosis, variable necrosis of the submucosa, but intact muscularis mucosae;**Grade 4**: Complete necrosis of both the mucosa and the submucosa, and loss of the muscularis mucosae;**Grade 5**: In addition to the changes of grade 4 there are also circulatory disturbances of the inner part of the external muscular layer (lamina muscularis externa);**Grade 6**: Complete necrosis of all layers of the intestinal wall.


When classifying the microscopical findings, it was noted that ischemic changes were not always consistent in each section. Sometimes, the nature and the magnitude of the changes varied from one part of the section to another. In these cases, the section was classified according to the Swerdlow degree that dominated quantitatively in the section. This semi-quantitative approach naturally involved some subjectivity. There were, however, no problems in achieving consensus between the two pathologists.

## 3. Results

Figure 2 shows the averaged SNV corrected diffuse reflectance spectra for each of the four cases with different intestinal tissue conditions. Clear differences between ischemic and reperfused tissue were observed on the reflectance spectra. The reflectance peak of oxyhemoglobin has a double absorption feature around 546 nm and 578 nm (Figure 2a,c,d), whereas deoxyhemoglobin for the ischemic cases has a single absorption feature around 563 nm (Figure 2b–d). The average reflectance intensity was higher for reperfused intestinal segments compared to ischemic segments in the wavelength range of 600–800 nm; this behaviour was reversed in the wavelength range 800–1000 nm as shown in Figure 2c,d. This observation is consistent with the reported absorption spectra of oxyhemoglobin and deoxyhemoglobin [23,24].

A zoomed-in plot of the diffuse reflectance spectra in the range of 500–600 nm (Figure 3a) makes it easier to observe where the changes related to oxyhemoglobin and deoxyhemoglobin levels occur. To visualize the spectral differences between data from viable and non-viable segments, Figure 3b shows the SNV transformed spectra from all measurement data (10 pigs) divided into two classes; red curves indicate segments histopathologically classified as viable and blue curves indicate segments classified as non-viable.

Figure 4 shows the relative change (relative to the reference Control Case 1) in the first derivative data for Cases 2, 3 and 4. From Figure 4b,c, an inversion of the curves was observed between data from ischemic segments and reperfused segments at several wavelengths. The largest difference between data from ischemia and reperfusion phase occurred in the wavelength range of 500–600 nm, where the difference between oxyhemoglobin and deoxyhemoglobin is most profound. The PLS coefficient (Figure 4d) reveals that the main contributions come from regions in the range of 500–600 nm and 750–800 nm. The PLS coefficients for the two- and three-class models were similar in the 500–600 nm region, but were less similar for the longer wavelengths.

Table 2a,b show the results from three- and two-class classification with PLS-DA models, respectively. Average scores (accuracy, sensitivity and specificity) with standard deviation from 10-fold cross-validation are reported.

The PLS-DA model was more accurate when classifying control, ischemic and reperfused intestinal segments, than when classifying viable and non-viable segments. Average accuracies of 97.8% and 92.5% respectively were achieved when using the SNV transformed data for three- and two-class classification models, respectively. The accuracies were higher when the first derivative data was used, for both two- and three-class classification.

To further test how accurately the PLS-DA model performs when there is extensive overlap of the spectra, two additional models were created to classify viable versus non-viable for data from the ischemia and reperfusion phases, respectively. The challenge is that spectra within the ischemia phase can be similar to each other and that the same is true for spectra within the reperfusion phase. Average accuracies of 82.5% and 82.8% were achieved when classifying viable versus non-viable, using the SNV transformed dataset and the first-order differentiated dataset from the ischemia phase. Average accuracies of 97.0% and 99.3% were achieved in the classification of viable versus non-viable, using the SNV transformed dataset and the first order differentiated dataset from the reperfusion phase.

### Histopathology

The microscopical investigations revealed that, with an increasing duration of ischemia, the damage to the wall of the small intestine was progressively higher. The morphological changes started in the mucosa, first with damage and loss of the surface epithelium and thereafter with increasing loss of both epithelium and connective tissue stroma. After 2 h of ischemia, many basal parts of mucosal glands were still intact, and the intestine wall was judged to be still viable at this stage (Figure 5b). After 3 h or more of ischemia, the damage to the mucosa was extensive, and at these stages the intestine was considered to be non-viable (Figure 6a,b). After 4 h of ischemia, there were large areas with complete mucosal and submucosal necrosis (Figure 6b).

## 4. Discussion

Diffusely reflected light contains information about the chemical composition of the sample (absorption) and the microstructure (scattering). For biological samples, the scattering properties are excessively complex [21], making it challenging to interpret the simultaneously occurring variations in the acquired spectra. In this study, important differences were observed in the spectra in the wavelength regions where oxyhemoglobin and deoxyhemoglobin influence absorption. The full spectral wavelength range was used to classify whether small intestinal segments were viable or non-viable, to investigate the accuracy and estimate clinical relevance.

VIS-NIR spectrometry can be compared to the bioimpedance method, where, instead of illuminating with and detecting light across a range of wavelengths, an electrical signal is applied across a range of frequencies, and the passive electrical properties of the tissue are measured. Low-frequency electrical bioimpedance (Hz and kHz range) mainly reflects the cellular structure [25], while bioimpedance in the MHz and GHz range depends more on the dipolar nature of biomolecules [3,26]. However, optical spectroscopy has an advantage over bioimpedance, in that measurements can be non-contact (for instance interactance measurements [27]). Non-contact bioimpedance measurements based on the induction of Eddy currents have been reported [28]. However, in most practical bioimpedance setups, a direct application of two or four electrodes is required [29].

A typical pulse oximeter uses two small light-emitting diodes, red (660 nm) and infrared (940 nm), to measure a photoplethysmography signal. This signal contains AC and DC components and is based on changes in reflectance due to pulses of arterial blood in the tissue. These components are used to calculate the oxygen saturation of the tissue. This method gives relative measurements as it uses the ratio between AC_660nm_/DC_660nm_ and AC_940nm_/DC_940nm_. However, ischemic intestine or recently reperfused intestinal tissue often has no pulse or a very weak pulse. This is a condition where a typical pulse oximeter will not measure any AC component and thus cannot give any estimation of oxygen saturation in the tissue. Diffuse reflectance spectrometry does not depend on a detectable pulse but provides a distinct reflectance curve over the measured wavelengths. This enables us to study the condition of the small intestine even when the arterial pulse is absent during ischemia. The VIS-NIR spectrum includes specific wavelengths that contain information about oxygen saturation and wavelengths that contain complex information about other chemical compositions and tissue microstructure.

In VIS-NIR spectroscopy, it is often hard to predict in advance which wavelength bands contain the most information relative to the analyte to be measured. Measuring the full wavelength range during a study enables the selection of the most relevant bands associated with ischemia–reperfusion injury.

There is a clear distinction between the control, ischemia, and reperfusion cases in Figure 2. For the control case (Figure 2a), variations during the 8-hour experiment are minor, indicating that there are minor changes in reflectance in the control segment, which is expected. For the ischemic and the reperfusion cases (Figure 2b–d), the differences compared to the control segments increase with ischemic time. Changes in the reflectance spectra during the ischemia and reperfusion period are not as obvious from one hour to another. There is a larger variation at 6, 7, and 8 h for Cases 2 and 4 (Figure 2b,d).

When ischemia is followed by reperfusion, the concentrations of oxyhemoglobin and deoxyhemoglobin change drastically. Figure 3a shows how the reflectance spectra in the wavelength range 500–600 nm switch from the ischemic to the reperfused state in Case 4 (4 h of ischemia followed by 4 h of reperfusion). The 500–600 nm region is dominated by differences between oxyhemoglobin and deoxyhemoglobin, which is a prominent feature related to ischemia–reperfusion injury. During ischemia, the level of deoxyhemoglobin increases as the tissue does not receive enough oxygenated blood, until the oxygen is entirely depleted. During reperfusion, there is an increase in oxyhemoglobin, as the tissue is perfused with oxygenated blood. This reflectance curve shape is consistent with the results reported by Karakas et al. [5], where similar spectral changes were observed for ischemic and perfused/reperfused intestinal segments (They plotted absorption instead of reflection in their figures).

Karakas et al. [5] conducted an experiment that was similar to ours, but focused on the ratio of the absorption between wavelengths at 560 nm and at 577 nm. We were not able to reproduce the associations between the ratio value to the histopathological grading as they pointed out in their study. The reason may be that we have used a different animal model and that the field of view of the optical fiber probe used in their study was considerably smaller than the probe used in this study. Sowa et al. [1] built a PLS-DA model on the spectroscopic data from the control and the ischemic segment and tested the model on data from reperfused segments. An average accuracy of 89% and sensitivity and specificity of over 80% were achieved in detecting ischemic tissue, but they did not classify viability.

The immunological responses during ischemia–reperfusion can increase the luminal epithelial permeability. After revascularization, increased vascular permeability will typically result in the formation of edema in the intestinal layers, especially the submucosa [30,31,32]. This can result in both chemical and microstructural changes that might explain some of the differences observed across the wavelength range.

When comparing the relative change between the first derivative of the intervention cases and the first derivative of the reference control case (Figure 4a–c), a partial inversion of the trace curve is observed. The trace curve crosses the zero line several times at different wavelengths, dependent on whether the case is ischemia or reperfusion. This reveals clear distinctions in the reflectance of VIS-NIR between the different cases. At local minima or maxima in the reflectance spectra, the first derivative curve crosses the zero line (on the *y*-axis). Whenever the spectra of the first derivative of the intervention groups (Cases 2, 3 and 4) overlap with the spectra of the first derivative of the reference case, the spectra in Figure 4a–c would cross the zero-line (on *y*-axis). As the control is the zero line in the plots, the magnitude of the deviation from the zero-line shows the wavelength dependent differences between the cases clearer than when plotting the raw data.

The PLS coefficients (Figure 4d) were quite similar for the two- and three-class models, with some differences around 760 nm, one of the oxyhemoglobin’s absorption maxima. This can indicate that the wavelength region 500–600 nm is equally important for both classification tasks. The regression coefficients in the region 750–800 nm were positively correlated with the shorter wavelength weights for the 3-class model and negatively correlated with the shorter wavelength weights for the 2-class model. This seems to follow the general shape of the oxyhemoglobin and deoxyhemoglobin absorption curve. In general, the regression vector places emphasis on the regions where the absorption maxima for both oxyhemoglobin and deoxyhemoglobin are present.

From the PLS coefficients (Figure 4d), we obtain an indication of the critical wavelengths for classifying viable and non-viable intestinal segments, and between the segments in the control, ischemia and reperfusion phases. This can help us to understand the potential for making a specialised instrument for this application. The wavelength region of 500–600 nm appears to be the most important for the models, and it may be sufficient to focus on these when designing a specialised instrument. Two models were built using only data from 500–600 nm; for two-class classification, mean accuracies of 94.0% and 92.0% were achieved with SNV transformed data and first-order differentiated data, respectively. For three-class classification, mean accuracies of 96.6% and 96.5% were achieved with SNV transformed data and first-order differentiated data. However, it is not conclusive whether including or excluding the region above 700 nm could make the model more or less robust; this would require a larger dataset.

It can be challenging to assess the blood flow condition in intestinal tissue based on visual inspection alone, as small differences in color can be difficult for the human eye to perceive. With reflectance spectroscopy and PLS-DA, it was possible to distinguish intestinal tissues with different injury conditions more accurately than the standard clinical method, where the success rate for identifying the non-viable segments during the first laparotomy is reportedly around 50% [30,33].

Regarding the generalizability of the classification models, 10-fold cross-validation was used, where each fold contained spectral data from one pig. During cross-validation, data from 9 pigs were used for training, and data from one pig were used for testing, ensuring that data from the same pig did not appear in both the training and test datasets. In this way, more generalized models were obtained, and data leakage was avoided, resulting in a more reliable model. It is unknown how transferable these results obtained from pig intestine are with respect to the human intestine, as diffuse reflectance spectroscopy on the human intestine has yet to be investigated in a similar setting. However, the pig intestine has important physiological and anatomical similarities to humans [34], and the mechanisms of ischemia–reperfusion injury are similar to the mechanisms observed in the human intestine [35]. Therefore, utilizing pig models to investigate ischemia–reperfusion injury can provide important insights and knowledge which may be transferable to the assessment of intestinal viability in humans.

The borderline between viable and non-viable intestine tissue was investigated by histopathological analysis. Earlier studies indicated that the possible upper limit for viability in the porcine mesenteric occlusion model is ≤3 h ischemic time [14]. In this study, the upper limit for viability seems to be ≤2 h of ischemic time.

The limited size of the biopsy sample and the heterogeneous composition of the lesions makes it challenging to evaluate the small intestine viability based on histological assessment [36,37]. Different injury degrees can be found within the same intestinal sample, and it is difficult to know whether changes seen in small samples are representative of the entire intestine. It is also difficult to speculate about how much remaining epithelium and muscle tissue is necessary for the intestine to be viable and functional.

The ’point of no return’ is hard to define and the time window for successful intervention is much harder to assess for small intestines when compared to the brain, heart, or kidney [30]. Current clinical preoperative assessment of intestinal viability after reperfusion can be challenging and, in many cases, a second look surgery after 24–72 h is required. During this period, the patient is at a high risk of acquiring intra-abdominal infections and sepsis. In severe cases, multi-organ failure and death occur. This further highlights the need for the development of decision support technologies that can assist the surgeon to resect only intestines that will not survive after reperfusion. Objective and accurate decision tools can help prevent short bowel syndrome, reduce the need for second-look surgery and ensure that locations selected for anastomosis are viable. This will improve the prognosis of the patients significantly.

The model used in this study only investigates diffuse reflectance spectroscopy during warm segmental full ischemia and does not investigate how intestinal tissue reflectance will change during other pathological developments. Mechanisms such as arterial/venous thrombosis and superior mesenteric artery occlusion may result in different spectral variations and signatures.

An ideal method to predict bowel viability after intestinal ischemia–reperfusion injury would offer the following features: high safety and easy handling, high accuracy, objectivity, reproducibility, and cost efficiency [38]. Spectroscopy measurements have the advantage of quick acquisition time, acquiring a spectrum from intestinal tissue takes <2 s. The DR probe used covers 24 mm × 47 mm of the small intestinal surface and allows the surgeon to scan areas of interest in a short time. As this method measures reflected light from the measurement site, it is not very dependent on the thickness of the tissue, unless the tissue becomes transparent. As the technique only applies light to the object being measured without applying any active signal that might pose a risk to the patient, and it does not require direct contact with the tissue (there is a requirement for isolation from ambient room light, which might require tissue contact), the measurement can easily be performed in a sterile environment. This allows for simplification of the clinical approval process for medical devices. The sum of these advantages makes this method potentially very user-friendly and efficient for real-time intraoperative assessment of the injury level of ischemic intestinal tissues. In the future, we are planning to test the interactance measurement, which allows for better optical sampling in the depth direction. A probe could be mounted over the tissue, non-contact, and monitor the relative change throughout the surgery.

This study used ten pigs, with four cases of ischemia–reperfusion models in each pig, with hourly measurements on each case over a period of 8 h. To further investigate this method, it would be necessary to assess other common intestinal ischemia–reperfusion pathological scenarios and to test on human intestines. To be able to assess this approach on human patients, different sterilization methods are currently being evaluated, for instance using hydrogen peroxide on the DR probe itself and using a sterile camera drape around the communication wire.

## 5. Conclusions

Using diffuse reflectance spectroscopy together with supervised partial least squares-discriminant analysis, it was possible to distinguish between normal, ischemic, and reperfused segments with an average accuracy of 99.2%. Furthermore, viable and non-viable segments were successfully classified with an average accuracy of 96.0%, which suggests that this approach has the potential to detect and assess the degree of ischemia–reperfusion injury non-invasively. Given that the measurement time required is around one to two seconds, this method can potentially be used to make real-time predictions and aid the decision-making process when estimating suitable margins for bowel resection during surgery. Histological analysis showed that the porcine intestinal segments are probably not viable after three hours of full ischemic occlusion. The initial results are very promising. Going forward, the method needs further refinement and validation in a larger study population and clinical validation on human patients with intestinal ischemia.

## Figures and Tables

**Figure 1 sensors-22-09111-f001:**
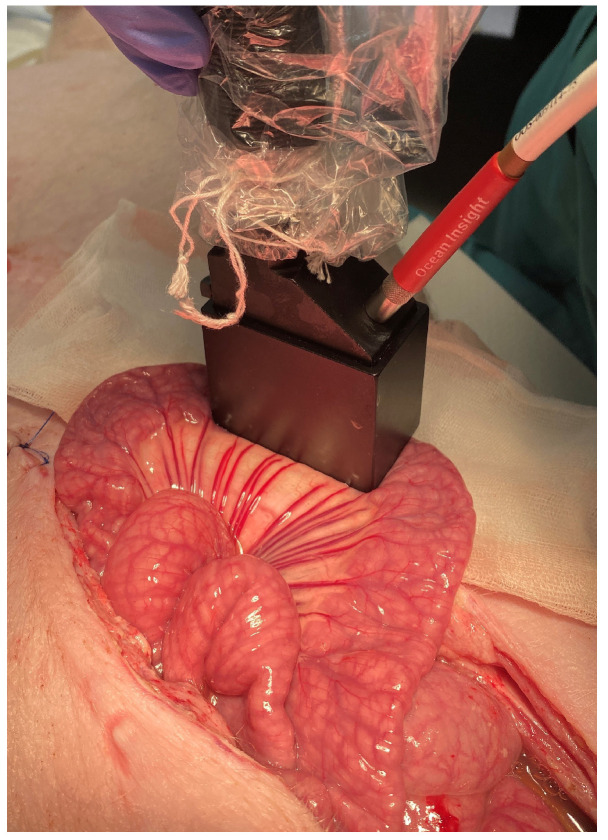
Experimental setup for in vivo measurements of a control small intestinal segment with the TC-DR probe.

**Figure 2 sensors-22-09111-f002:**
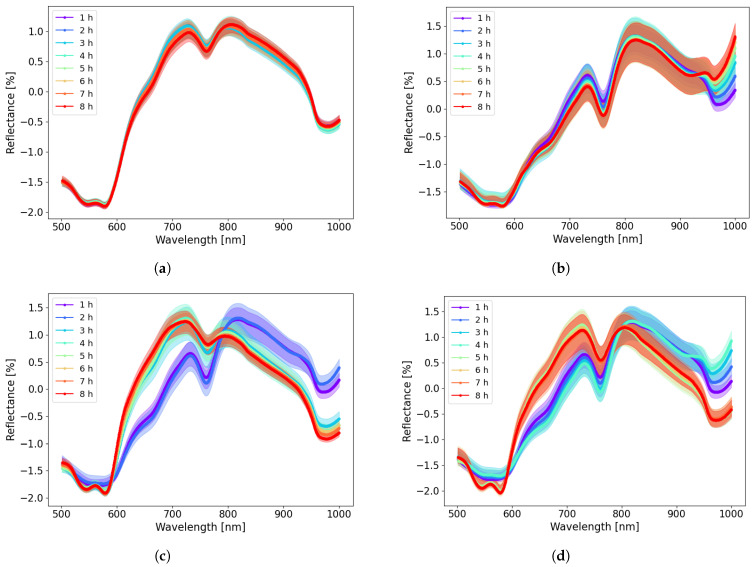
Standard normal variate corrected reflectance spectra for four cases as a function of wavelength using Equation (Equation 2). Mean and 95 % confidence intervals are shown in the figures. *N* = 10 for each of the groups. (**a**) Reflectance for Case 1 (Control—8 h of normal perfusion). (**b**) Reflectance for Case 2 (Ischemia—8 h of warm full ischemia). (**c**) Reflectance for Case 3 (2 h of ischemia followed by 6 h of reperfusion). (**d**) Reflectance for Case 4 (4 h of ischemia followed by 4 h of reperfusion).

**Figure 3 sensors-22-09111-f003:**
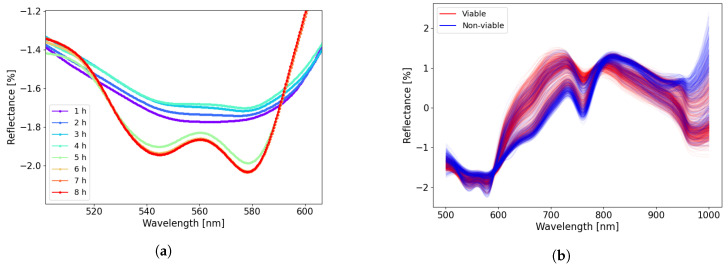
(**a**) A zoomed-in plot in region 500–600 nm of the SNV transformed reflectance data for Case 4 (4 h of ischemia followed by 4 h of reperfusion). (**b**) All acquired spectra separated into viable and non-viable classes.

**Figure 4 sensors-22-09111-f004:**
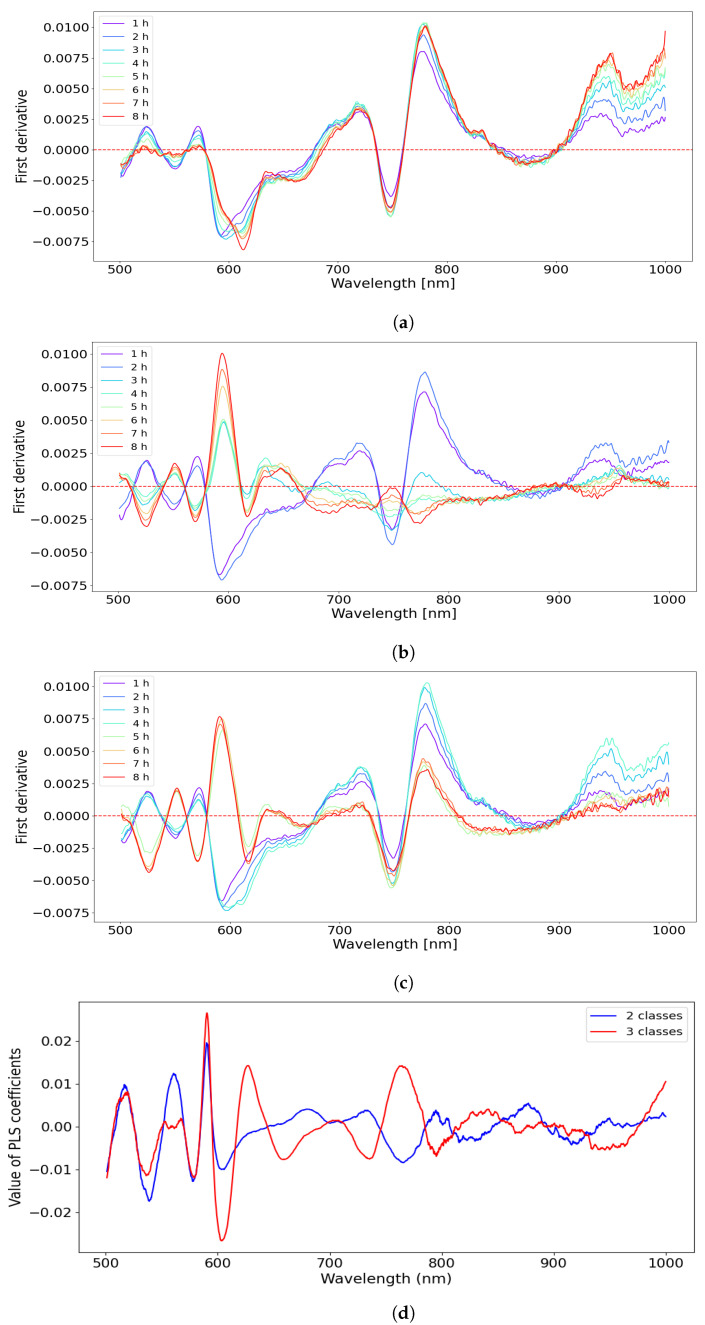
(**a**–**c**) The difference in the first derivative of reflectance data between Cases 2, 3, 4 and the control Case 1. (**d**) The value of PLS coefficients as a function of wavelength for both two- and three-class models.

**Figure 5 sensors-22-09111-f005:**
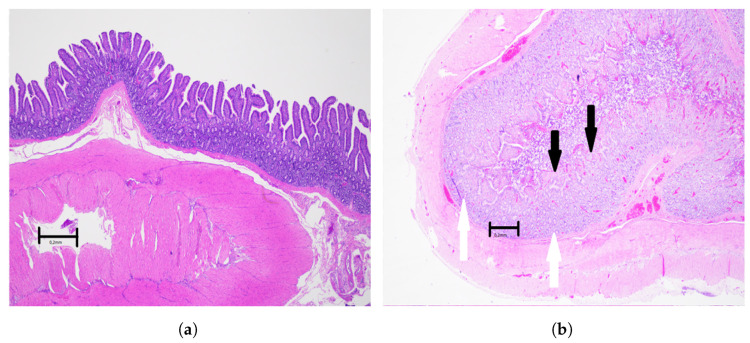
Light microscopical pictures of selected structures of the viable small intestine. Objective magnification ×2. Scale bar indicates 0.2 mm. (**a**) Light microscopical picture of the wall of the healthy control small intestine, with no injury, photographed before the start of ischemia. All layers of the intestinal wall are intact and without pathological changes, i.e., modified Swerdlow grade 0. (**b**) Light microscopical picture of the wall of the small intestine after 2 h of ischemia followed by 2 h of reperfusion. There is extensive damage to the surface epithelium of the mucosa (black arrows) and areas of substance loss in the mucosal lamina propria. The lamina muscularis mucosae is preserved, and some basal parts of glands are intact (white arrows), classified as modified Swerdlow grade 2.

**Figure 6 sensors-22-09111-f006:**
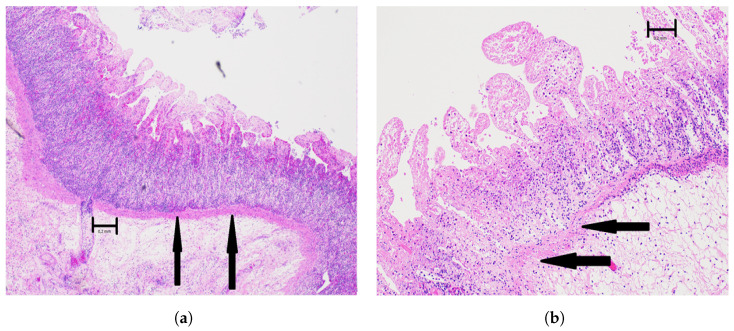
Light microscopical picture of selected structures of the non-viable small intestine. Objective magnification ×2. Scale bar indicates 0.2 mm. (**a**) A light microscopical picture of the small intestine after 3 h of ischemia and 5 h of reperfusion. There is complete mucosal necrosis with loss of villus epithelium, and basal parts of the glands are not preserved. There is also edema and other circulatory disturbances of the submucosa. The lamina muscularis mucosa is intact (black arrows), classified as modified Swerdlow grade 3. (**b**) The light microscopical picture of the small intestinal wall after 4 h of ischemia followed by 4 h of reperfusion. There is complete necrosis of the mucosa with no remaining viable epithelium. The submucosa shows complete necrosis with edema and dilated vessels. The lamina muscularis mucosae is fragmented and destroyed to about one-third of its length in this picture (black arrows), classified as modified Swerdlow grade 4.

**Table 1 sensors-22-09111-t001:** Overview of cases and viability. The white background indicates control with normal perfusion, the blue background indicates ischemia, and the red background indicates reperfusion.

Hours	Case 1	Case 2	Case 3	Case 4
**1–2**	Viable	Viable	Viable	Viable
**3–4**	Viable	Non-viable	Viable	Non-viable
**5–6**	Viable	Non-viable	Viable	Non-viable
**7–8**	Viable	Non-viable	Viable	Non-viable

**Table 2 sensors-22-09111-t002:** Classification results of 10-fold cross-validation using PLS-DA with mean and standard deviation. Two different datasets were used, one dataset was SNV transformed, and the other dataset was SNV transformed and first-order differentiated. (**a**) Three-class classification between control, ischemic and reperfused intestinal segments. (**b**) Two-class classification between viable and non-viable intestinal segments.

a. Three-class model classification results			
Control VS ischemia VS Reperfusion			
Data	Accuracy [%]	Sensitivity [%]	Specificity [%]
SNV	97.8 ± 1.3	99.1 ± 1.3	97.8 ± 3.1
SNV Deri	99.2 ± 0.9	99.5 ± 0.8	98.9 ± 1.9
**b. Two-class model classification results**			
Viable VS Non-viable			
Data	Accuracy [%]	Sensitivity [%]	Specificity [%]
SNV	92.5 ± 2.9	87.8 ± 11.3	95.3 ± 4.4
SNV Deri	96.0 ± 2.8	94.2 ± 6.5	97.2 ± 2.6

## Data Availability

The data presented in this study are available on request from the corresponding author.

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
