# Peer review of "Assessment of Intestinal Ischemia–Reperfusion Injury Using Diffuse Reflectance VIS-NIR Spectroscopy and Histology"

_sensors, 2022, doi:10.3390/s22239111_

Round 1

Reviewer 1 Report

In this manuscript, the authors extended the application of VIS-NIR diffuse reflectance spectroscopy into the studies of ischemia-reperfusion injury in the small intestine of pigs. The models they built could be used to differentiate between ischemic and reperfused intestinal segments, as well as between viable and non-viable parts with very high accuracy. Their work could reduce the need for second-look surgery and improve the decision-making process. I feel the manuscript meets the scope of Sensors, and below are some comments.

Suggestions to authors for improving the manuscript:

1. Each spectrum is the average of signals from a small area. What’s the size of the detection area, i.e., the field of view, in the optical setup?

2. What’s the definition of accuracy, sensitivity, and specify, respectively, in Table 2?

3. In Table 2(b), there is a larger fluctuation in sensitivity. What’s the reason for this?

4. Compared to SNV-corrected data in Table 2, why does SNV derivative data exhibit better performance?

5. In Figure 4(a), (b), and (c), the legend blocks a part of the plots.

6. What’s the field of view in Figures 5 and 6? Add scale bars to indicate the size of the image.

Reviewer 2 Report

A porcine model was used to investigate the feasibility of using VIS-NIR spectroscopy to differentiate between degrees of ischemia-reperfusion injury in small intestine. Ten pigs were used in this study and four segments were created in the small intestine of each pig: (1) control, (2) full arterial and venous mesenteric occlusion for 8 hours, (3) arterial and venous mesenteric occlusion for 2 hours followed by reperfusion for 6 hours, and (4) arterial and venous mesenteric occlusion for 4 hours followed by reperfusion for 4 hours. Two models were built using partial least square discriminant analysis. The first model was able to differentiate between the control, ischemic and reperfused intestinal segments with an average accuracy of 99.2 % with 10-fold cross-validation, and the second model was able to discriminate between the viable versus non-viable intestinal segments with an average accuracy of 96.0 % using 10-fold cross-validation. Moreover, histopathology was used to investigate the borderline between viable and non-viable intestinal segments. The VIS-NIR spectroscopy method together with a PLS-DA model showed promising results and appears to be well-suited as a potentially real-time intraoperative method for assessing intestinal ischemia-reperfusion injury, due to its easy-to-use and non-invasive nature. My main concerns about this manuscript are 1) the novelty/scientific contributions are not clear 2) the results and comparison section is weak, the comparison with state-of-the-art research is missing. Therefore, authors are advised to resubmit this manuscript after addressing this concern.
